# Differences in Risk Factors for Diabetic Retinopathy in Type 1 and Type 2 Diabetes Mellitus Patients in North-East Poland

**DOI:** 10.3390/medicina56040177

**Published:** 2020-04-14

**Authors:** Wojciech Matuszewski, Magdalena M. Stefanowicz-Rutkowska, Magdalena Szychlińska, Elżbieta Bandurska-Stankiewicz

**Affiliations:** Clinic of Endocrinology, Diabetology and Internal Medicine, Department of Internal Medicine, School of Medicine, Collegium Medicum, University of Warmia and Mazury, 10-561 Olsztyn, Poland; M.M.Stefanowicz@gmail.com (M.M.S.-R.); magdaaa4@o2.pl (M.S.); bandurska.endo@gmail.com (E.B.-S.)

**Keywords:** diabetes mellitus, diabetic retinopathy, risk factors

## Abstract

*Background and Objective*: Nowadays, diabetes is one of the main causes of blindness in the world. Identification and differentiation of risk factors for diabetic retinopathy depending on the type of diabetes gives us the opportunity to fight and prevent this complication. Aim of the research: To assess differences in the risk factors for diabetic retinopathy in type 1 and type 2 diabetes mellitus patients in Warmia and Mazury Region, Poland. *Materials and Methods*: Risk factors for diabetic retinopathy (DR) were assessed on the basis of an original questionnaire, which included: personal data, clinical history of diabetes and eye disease. Elements of clinical examination: blood pressure, BMI, waist circumference. Indicators of diabetes metabolic control: mean glycemia, glycated hemoglobin (HbA1c), total cholesterol and triglycerides, creatinine, glomerular filtration rate (GFR), albumin–creatinine ratio in urine. *Results*: The study group included 315 (26%) patients with DM1 and 894 (74%) patients with DM2. Risk factors were estimated on the basis of logistic regression and verified with Student’s t-test. Statistically significant dependencies were found in both groups between the occurrence of diabetic retinopathy and diabetes duration, HbA1c, triglyceride concentrations, indicators of kidney function and cigarette smoking status. In the DM2 group, the development of DR was significantly influenced by the implemented models of diabetic treatment. *Conclusions*: In the whole study group, the risk of DR was associated with the duration of diabetes, HbA1c, triglyceride concentrations and smoking. In DM1 patients, the risk of DR was associated with diabetic kidney disease in the G1A1/A2 stage of chronic kidney disease, and in DM2 patients with the G2 stage of chronic kidney disease. An important risk factor for DR in DM2 patients was associated with late introduction of insulin therapy.

## 1. Introduction

Diabetes has been known about for thousands of years. Lesions in the fundus of the eye were first found in the course of diabetes in 1855 by Viennese ophthalmologist Eduard von Jaeger, who termed them ‘retinitis diabetica’ [1,2]. At present, we are witnessing a constant increase in the prevalence of diabetes and longer life expectancy of diabetic patients, which leads to the development of a number of microvascular and macrovascular complications, including diabetic retinopathy [3,4,5]. The strongest prognostic factor for the development and progression of diabetic retinopathy (DR) is its duration [6,7,8]. In turn, the most significant modifiable factor for the development of DR consists of hyperglycemia, which triggers cascade lesions in polyol pathway, PKC activation and non-enzymatic protein glycation. This factor also leads to increased oxidation stress and increases synthesis of prothrombotic factors, in turn contributing to the origination of areas of oxygen deficiency and stimulating increased production of vascular endothelial growth factor (VEGF). VEGF is responsible for the development of pathological angiogenesis and proliferative diabetic retinopathy, leading to eyesight loss [9,10]. Other recognized risk factors for DR include: lipid metabolism disorders, hypertension, diabetic kidney disease, pregnancy, puberty, cataract surgery or history of kidney–pancreas or only kidney transplant [11,12]. In recent decades a number of publications assessing risk factors for DR for only one DM type [13] have appeared, yet only a few of them have analyzed dependence on diabetes type. The aim of this study was thus to determine particular risk factors for DR and assess their correspondence to type 1 and 2 diabetes.

## 2. Materials and Methods

### 2.1. Study Population

The study of risk factors for DR presented in this paper was conducted in 2012–2016 in the Warmian-Masurian Voivodship in north-east Poland. The prevalence rate of diabetes in Warmia and Mazury according to the data collected by the National Health Fund is 3,5% [14]. The study population consisted of adult type 1 and 2 diabetic patients diagnosed according to WHO criteria [15]. The study encompassed patients hospitalized (inpatients) in the Endocrinology, Diabetology and Internal Diseases Clinic of the University of Warmia and Mazury. Outpatients who were taken care of by specialist diabetes centers and family doctors were randomly selected from the area of the voivodship. The study was approved (approval number 10/2010, date 25.03.2010) by the the Bioethics Committee of the Faculty of Medicine University of Warmia and Mazury in Olsztyn, Poland. 

### 2.2. Methods of Assessing Risk Factors for Diabetic Retinopathy

Risk factors for DR were assessed on the basis of a specially designed questionnaire which included: personal data, medical history of diabetes and ophthalmological problems (eye fundus) as well as elements of physical examination: blood pressure, BMI and waist circumference. Hypertension was defined according to the European Society of Hypertension (ESH) and European Society of Cardiology (ESC) criteria, BMI was calculated by applying Quetelet’s equation, while the waist circumference measurement was interpreted according to the International Diabetes Federation (IDF) criteria [16,17,18]. Moreover, metabolic control indicators were assessed: mean daily glycemia, glycated hemoglobin (HbA1c), concentrations of total cholesterol and triglycerides, concentration of creatinine, glomerular filtration rate (GFR), urine albumin–creatinine ratio (Table 1).

### 2.3. Statistical Analysis

Statistical analysis was performed with the STATISTICA 10 PL statistics package (StatSoft Polska, Kraków, Poland) and the econometric GRETL 1.9.9 CVS program. When it came to comparing mean values in the study groups, hypotheses were tested with the use of the non-parametric Mann–Whitney U-test. The normality of distribution of variables in both interval and ratio scales was tested with the Shapiro–Wilk test, while homogeneity of variance was tested with Levene’s test. For analyzing the effect of variables in measurable scales or the effect of the ordinal scale on the dichotomous variable, the logit model was used, and also for assessing the method of the highest reliability and considering correction of heteroscedasticity of random elements of the model. McFadden’s Pseudo R-Squared was applied to evaluate the fit of the assessed regressions. The accuracy of the alignment was evaluated ex-post with count-R^2^. The significance of the logit model was tested with the χ^2^ likelihood ratio test. For analyzing relationships between variables in nominal scales or ordinal scales, contingency tables and the chi-squared test were used, while the strength and direction of relationships between variables in interval and ratio scales were checked with the Pearson correlation coefficient. The significance level *p* = 0.05 was determined for all the tests.

## 3. Results

The study encompassed 1209 diabetic patients, including 639 women (53%). There were 315 (26%) DM1 patients aged 37.0 (13.55) years old with diabetes duration of 12.3 (9.78) years; and there were 894 (74%) DM2 patients aged 61.2 (11.13) years old with diabetes duration of 10.5 (8.09) years. Sixty-one percent of patients were under family doctors’ care, while the remaining ones were under diabetologists’ supervision. Diabetic retinopathy was found in 25.48% of patients in the whole experimental group. DM1 patients were diagnosed with DR in 32.58% of cases and DM2 in 23.04% of cases. 

The characteristic features of the study groups and metabolic control indicators are presented in Table 2.

For the whole study population of DM patients, dependencies between percentage of HbA1c, concentration of triglycerides and waist circumference were found. A positive correlation between HbA1c, concentration of triglycerides and waist circumference was found. In DM1 patients, no strong correlation between particular risk factors for DR was determined. In DM2 patients, a positive correlation between HbA1c, concentration of triglycerides and waist circumference was found (Figure 1).

The analysis of mean values and standard deviations for the analyzed risk factors for DR with reference to its advancement showed that there were statistically significant differences for the percentage of HbA1c (*p* < 0.05) (Figure 2).

The analysis of the strength and direction of relationships between particular metabolic control indicators in DM1 and DM2 patients showed that there were a number of statistically significant correlations (Table 3).

The assessed values of structural parameters of logistic regressions showed that a number of factors had a significant influence on an increase in probability of DR in DM1 patients. These were: DM duration, albumin–creatinine ratio, GFR, HbA1c and cigarette smoking. In DM2 patients such factors included: DM duration, GFR, HbA1c and cigarette smoking.

A significant increase in the risk of DR was observed in DM1 patients with diabetic nephropathy already in the G1A1/A2 stage of chronic kidney disease, while in DM2 patients this occurred as late as in the G2 stage. Introducing combination therapy in DM2 patients significantly decreased the probability of DR (Table 4).

## 4. Discussion

The present paper confirms that the main, and at the same time unmodifiable, risk factor for DR is DM duration, irrespective of DM type. It was proved that hyperglycemia is a modifiable risk factor for DR, showing the relationship between HbA1c percentage and risk and advancement of DR in the whole study population. Kilpatrick et al. stated that long-term fluctuation of HbA1c results in a risk of DR in DM1 patients, while short-term fluctuation does not lead to the development of microangiopathy [19,20]. Hietela et al. observed that changeability of HbA1c is related to an increased risk of DR, requiring laser therapy in DM1 patients [21]. Sartore et al. confirmed by continuous glucose monitoring (CGM) that glucose variability, regardless of HbA1c, may also have a role as a risk factor for DR in patients with DM1 and DM2, particularly in the case of acute fluctuations and acute hyperglycemia [22]. However, in the group of DM2 patients, some authors have not observed a positive correlation between HbA1c and the development of DR [23].

In the present study, it was observed that a significantly increased risk of DR appeared in DM1 patients with hyperfiltration or normal glomerular filtration yet with an increased urine albumin–creatinine ratio, that is, in the G1A1/A2 stage of chronic kidney disease. In DM2 patients the risk of DR correlated significantly with lowered values of GFR, which showed that one of the risk factors for DR in DM2 patients was chronic kidney disease, yet as late as in the G2 stage.

The above result was confirmed in a study on a population of 947 DM2 patients, which proved that an increased excretion of albumins in urine was related to a higher prevalence of DR, neuropathy and cardiovascular diseases. The authors suggest that urinary albumin excretion may reflect a state of generalized vascular damage [24]. Kodali et al. confirmed that in patients with an albumin–creatinine ratio greater than 2 mg/mmol, retinopathy and diabetic neuropathy were observed significantly more often [25]. A Chinese study published in March 2015 demonstrated that a GFR value of ≤ 99.4 mL/min/1.73 m^2^ in DM2 patients may point to the presence of early-stage DR [26]. In another study, Chen et al. observed that in patients with microalbuminuria and GFR > 60 mL/min/1.73 m^2^ the risk of DR was three times higher than in patients with normal albuminuria and GFR 30–59.9 mL/min/1.73 m^2^ [27].

In patients with diabetic nephropathy manifesting with microalbuminuria or proteinuria, concentrations of lipoproteins and fibrinogen increase and disorders of platelet function occur. These disorders are related also to the development of DR [28,29]. Statistically significant correlations were shown in the the Renin-Angiotensin-System Study (RASS) study between DR and preclinical diabetic nephropathic lesions in DM1 patients [30]. In turn, the Wisconsin Epidemiologic Study of Diabetic Retinopathy (WESDR) confirmed that in diabetes lasting for 10 years proliferative diabetic retinopathy (PDR) was three times more frequent in patients with proteinuria. According to some scholars, diabetic nephropathy has always been accompanied by PDR, while albuminuria and proteinuria have been important independent predicators of development and progression of DR [31,32,33,34].

In studies conducted on DM patients, it was demonstrated that cholesterol concentration was an independent risk factor for nephropathy [35]. When it comes to DR, authors claimed that the effect of this factor was not so obvious [36,37]. It was shown that in patients treated with insulin, higher cholesterol concentrations were related to an increased occurrence of hard exudates [34]. In the Early Treatment Diabetic Retinopathy Study (ETDRS) it was confirmed that every increase in cholesterol concentration by 10% increased the number of hard exudates by 50% in older DM1 patients [37,38]. In subsequent studies it was also found that there were positive correlations between cholesterol concentration, triglycerides concentration and the occurrence of hard exudates, whose number did not decrease after photocoagulation, but did after statin therapy [39,40]. At the same time, fenofibrate therapy lowered the need for laser photocoagulation in diabetic patients, which was confirmed by the FIELD study results [41,42]. In the Diabetes Control and Complications Trial (DCCT) study conducted on 998 DM1 patients, it was shown that the degree of advancement of DR was related to an increase in triglycerides concentration and a decrease in HDL-cholesterol concentration [43]. A Chinese study of DM2 patients proved that an increased VLDL (Very Low Density Lipoprotein) and TG concentrations were independent factors for an increased risk of DR [44]. In addition, a non-traditional lipid marker such as Apo might be a candidate for better prediction of DR severity than traditional lipid markers [45]. The presented data are in line with the results of this study, with an indirect risk factor for DR in DM1 and DM2 patients being an increased concentration of triglycerides.

In the present study, BMI did not increase the probability of DR. However, Kaštelan et al. investigated the effect of BMI on progression of DR. Numerous scholars have concluded that DR is related to BMI only in its correlation with HbA1c, cholesterol concentration and hypertension [46]. Price postulated that obesity in DM1 patients may lead to the progression of retinopathy and macrovascular complications [47].

The present study also shows that cigarette smoking significantly increased the probability of DR in the whole study population. Similar conclusions were proffered by Uruska and Muhlhauser et al. with reference to DM1 patients [48,49]. Other authors also confirmed an unfavorable relationship between cigarette smoking and the occurrence and progression of DR [50,51,52]. This thesis was not supported by Stratton et al., and in the UK Prospective Diabetes Study (UKPDS) study it was shown that, in DM2 patients, progression of DR was related to non-smoking [53,54].

Recent years have witnessed numerous publications on the effect of treatment programmes and reduction of particular risk factors for DR on the subsequent occurrence of this complication. Yet there are very few reports concerning direct relationships between the manner of treating diabetes and the occurrence of DR. In a study conducted in Kuwait on a group of 165 DM2 patients, it was proved that treatment with sulfonyl–urea derivatives or insulin in patients who did not achieve good metabolic control was an independent statistically significant factor related to the occurrence of DR [55]. The present study demonstrated that too slow intensification of antidiabetic treatment in DM2 patients, that is, introducing insulin therapy too late, was an independent risk factor that accelerated the development of DR. This observation confirmed results of such studies as the UKPDS, DCCT, EUCLID, Kumamoto Study or Wisconsin Epidemiologic Study. These studies all demonstrated that optimizing control of glycemia through intensifying antidiabetic treatment decreases the frequency of microangiopathic complications [54,56,57,58,59,60].

## 5. Conclusions

Diabetes is an epidemic of the modern world, having a history of long-term consequences that have brought about human suffering and economic costs. Diabetic retinopathy is the most serious complication in the eye caused by diabetes; it can occur as early as at the moment of diagnosing the disease, and contributes to 80% of the causes of eyesight loss in the diabetic population. Thorough knowledge about different risk factors for DR is essential in order to prepare and implement prevention programs and limit the progression of this serious diabetic complication. The current study, as well as other similar studies, need to be continued not only because of their academic value, but also because of their applicability in everyday medical practice and the fact that one of the greatest fears of patients is eyesight loss.

## Figures and Tables

**Figure 1 medicina-56-00177-f001:**
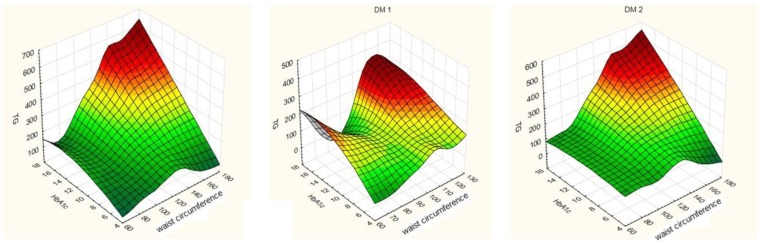
Correlation between percentage of glycated hemoglobin (HbA1c), concentration of triglycerides and waist circumference in diabetes mellitus (DM) DM1 and DM2 patients.

**Figure 2 medicina-56-00177-f002:**
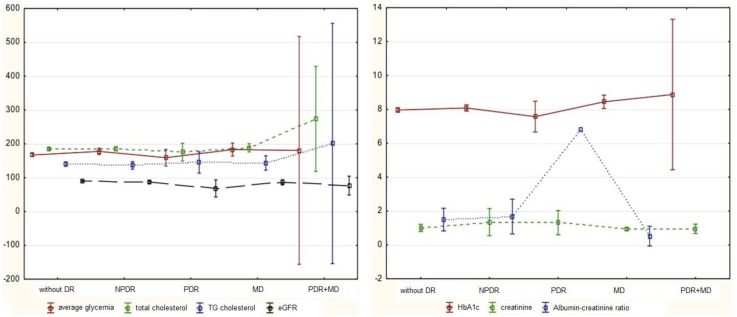
Mean values and standard deviations for the assessed indicators with reference to the advancement of diabetic retinopathy.

**Table 1 medicina-56-00177-t001:** Normal and recommended ranges of assessed indicators of metabolic balance in diabetes.

Metabolic Balance Indicator	Recommended Range
Glycated hemoglobin (HbA1c)	≤7.0% (≤53 mmol/mol)
Concentration of total cholesterol	<175 mg/dL (<4.5 mmol/L)
Concentration of triglycerides	<150 mg/dL (<1.7 mmol/L)
Concentration of creatinine	0.5–0.9 mg/dL
Glomerular filtration rate (GFR)	>60 mL/min
Urine albumin–creatinine ratio	0.0–2.5 mg/mmol

**Table 2 medicina-56-00177-t002:** Characteristics of the study group with metabolic control indicators. OAD—oral anti-diabetes drugs; IT—insulin therapy; TG—triglycerides, ACR—albumin to creatinine ratio, BP—blood pressure, BMI—body mass index.

	Study Group						
	Total	DM1	DM2	*p*	With DR	Without DR	*p*
***n* (%)**	1209 (100)	315 (26)	894 (74)	-	308 (25.48)	901 (74.52)	-
**Age—years (SD)**	54.9 (15.79)	37.04 (13.55)	61.2 (11.13)	0.000	56.1 (14.18)	54.5 (16.30)	0.451
**Diabetes duration—years (SD)**	11.0 (8.60)	12.3 (9.78)	10.5 (8.09)	0.038	17.4 (8.72)	8.8 (7.39)	0.000
**OAD (%)**	351 (29)	0 (0)	351 (39)	1.000	39 (13)	312 (35)	0.005
**IT (%)**	520 (43)	315 (100)	205 (23)	1.000	161 (53)	359 (40)	0.005
**OAD + IT (%)**	338 (28)	0 (0)	338 (38)	1.000	108 (34)	230 (25)	0.089
**HbA_1c_ (%)**	8.00 (1.79)	8.44 (2.02)	7.84 (1.67)	0.000	8.1 (1.42)	8.0 (1.90)	0.000
**Total cholesterol (mg/dL)**	184.64 (44.14)	182.39 (39.32)	185.42 (45.70)	0.683	186.6 (45.1)	183.9 (43.79)	0.493
**TG cholesterol (mg/dL)**	139.84 (83.66)	106.28 (63.97)	151.64 (86.56)	0.000	138.8 (75.74)	140.2 (86.42)	0.576
**Creatinine (mg/dL)**	1.07 (3.51)	0.85 (0.38)	1.15 (4.12)	0.000	1.2 (4.84)	1.0 (2.89)	0.117
**eGFR (mL/min/1.73 m^2^)**	88.81 (25.81)	101.10 (25.80)	84.04 (24.02)	0.000	85.7 (26.39)	90.0 (25.52)	0.029
**ACR (mg/mmol)**	1.54 (3.09)	0.53 (0.74)	2.08 (3.68)	0.019	1.7 (2.78)	1.5 (3.22)	0.353
**Systolic BP (mmHg)**	133.38(17.12)	127.70 (15.15)	135.39 (17.33)	0.000	135.6 (19.51)	132.6 (16.17)	0.040
**Diastolic BP (mmHg)**	78.90 (9.44)	77.54 (8.97)	79.38 (9.56)	0.015	78.3 (9.47)	79.1 (9.43)	0.128
**BMI (kg/m^2^)**	28.95 (6.09)	24.24 (4.00)	30.61 (5.82)	0.000	28.4 (5.69)	29.1 (6.21)	0.113
**Waist circumference (cm)**	98.75 (15.37)	87.06 (10.88)	102.35 (14.76)	0.000	98.1 (15.47)	99.0 (15.35)	0.557

**Table 3 medicina-56-00177-t003:** Correlation coefficients for particular metabolic control indicators in DM1 and DM2 patients (* the value of correlation coefficient shows a statistically significant correlation).

	**DM1 Patients**										
	**DM Duration**	**BMI**	**Waist**	**Mean Glycemia**	**HbA1c**	**Cholesterol**	**TG**	**Creatinine**	**GFR**	**Albumin–Creatinine Ratio**	**Systolic APB**
BMI	0.089	1.000									
Waist	−0.058	0.677 *	1.000								
Mean glycemia	0.051	−0.458	−0.509 *	1.000							
HbA1c	−0.164	−0.475	−0.360	0.772 *	1.000						
Cholesterol	0.132	−0.262	−0.477	0.390	0.193	1.000					
TG	−0.381	−0.084	−0.176	0.452	0.530 *	0.588 *	1.000				
Creatinine	−0.065	−0.034	0.185	−0.158	−0.164	−0.293	−0.268	1.000			
eGFR	−0.002	−0.087	−0.390	0.509 *	0.231	0.468	0.489	−0.625 *	1.000		
Albumin−creatinine ratio	0.145	−0.075	−0.270	−0.228	−0.159	0.120	−0.179	0.045	−0.149	1.000	
Systolic ABP	−0.079	−0.010	0.197	−0.176	−0.353	−0.200	−0.396	0.654 *	−0.171	0.136	1.000
Diastolic ABP	0.460	−0.197	0.039	0.222	0.338	−0.213	−0.240	0.472	−0.387	−0.017	0.236
	**DM2 patients**										
	**DM Duration**	**BMI**	**Waist**	**Mean Glycemia**	**HbA1c**	**Cholesterol**	**TG**	**Creatinine**	**GFR**	**Albumin–Creatinine Ratio**	**Systolic ABP**
BMI	0.001	1.000									
Waist	0.036	0.903 *	1.000								
Mean glycemia	−0.182	0.347 *	0.417 *	1.000							
HbA1c	−0.237	0.222	0.263	0.758 *	1.000						
Cholesterol	−0.154	−0.196	−0.204	−0.068	0.025	1.000					
TG	−0.108	0.370 *	0.398 *	0.461 *	0.439 *	0.340	1.000				
Creatinine	0.046	0.158	0.283	−0.110	0.021	−0.160	−0.090	1.000			
eGFR	−0.031	−0.201	−0.253	0.063	−0.022	0.244	−0.037	−0.837 *	1.000		
Albumin–creatinine ratio	0.014	0.281	0.357 *	0.362 *	0.106	0.035	0.351 *	0.051	−0.106	1.000	
Systolic ABP	−0.402 *	0.042	0.065	0.116	0.048	0.050	0.168	0.127	−0.074	0.183	1.000
Diastolic ABP	−0.584 *	−0.269	−0.229	0.026	0.044	0.063	0.177	−0.079	0.044	0.035	0.469 *

**Table 4 medicina-56-00177-t004:** Risk factors for DR in DM1 and DM2 patients: results of diagnostic tests in logistic models. (* risk factors with a significant impact on the development of diabetic retinopathy)

	**DM1 patients**					
**Model**	**DM Duration (SE)**	**Albumin–Creatinine Ratio (SE)**	**GFR (SE)**	**HbA1c (SE)**	**Cigarette Smoking (SE)**	
1	0.17 * (0.06)	1.06 * (0.61)	0.04 (0.02)	-	-	
2	0.25 * (0.09)	0.82 (0.73)	0.07 * (0.02)	-	22.16 * (1.69)	
3	0.16 * (0.06)	1.05 * (0.31)	0.04 (0.02)	-	-	
4	0.09 * (0.02)	-	0.01 * (0.008)	0.26 * (0.09)	-	
	**DM2 patients**					
**Model**	**DM Duration (SE)**	**Albumin–Creatinine Ratio (SE)**	**GFR (SE)**	**HbA1c (SE)**	**Cigarette Smoking (SE)**	**Treatment Model (SE)**
1	0.11 * (0.04)	0.05 (0.09)	−0.02 (0.02)	-	-	-
2	0.10 * (0.04)	0.04 (0.09)	−0.02 (0.02)	-	18.55 * (0.55)	-
3	0.14 * (0.06)	0.02 (0.11)	−0.03 (0.02)	-	-	−1.12 * (0.55)
4	0.09 * (0.02)	-	−0.02 * (0.01)	0.26 * (0.09)	-	-

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
