# Peer review of "Differences in Risk Factors for Diabetic Retinopathy in Type 1 and Type 2 Diabetes Mellitus Patients in North-East Poland"

_medicina, 2020, doi:10.3390/medicina56040177_

Round 1

Reviewer 1 Report

The paper is very interesting and based on relatively wide range of data. I am sure it will be cited many times as it provides also some epidemiological insights into diabetic population in Central European region. It is very valuable, as there is substantial lack of such data in this area. The idea to look for DM1 and DM2 DR risk factors separately is useful to determine how to approach those different groups of patients. However, Authors should mention that there are studies concerning DR risk factors for only one DM type (e.g. https://www.ncbi.nlm.nih.gov/pmc/articles/PMC6174939/).

Few remarks:

Literature – discussion is based on relatively old papers. Although cited large studies results are very important Authors should also look for current related data. When phrase “diabetic retinopathy risk factors” is searched in PubMed, there is a lot of important papers published in last 5 years.

Table 2 - English phrases in description should be corrected.

Author Response

Dear Dear Sir or Madam

Thank you so much for your valuable review. I corrected all comments. Thanks again for your time. Corrected manuscript attached. 

Yours faithfully

Wociech Matuszewski 

Reviewer 2 Report

Authors made a good attempt to explain the key differences in risk factors for DR type 1 and Type 2 DM patients in north-War Poland. There is one key question that need to be addressed.

  1. Authors should include some of the latest references in the introduction and discussion sections.

When working on these epidemiological studies latest studies should be included in the present context.

Author Response

(The authors gave the same response as above.)
